# Effects of Weak Bedding Plane, Fault, and Extreme Rainfall on the Landslide Event of a High Cut-Slope

**DOI:** 10.3390/s22186790

**Published:** 2022-09-08

**Authors:** Yiqing Sun, Deying Li, Fasheng Miao, Xiangjie She, Shuo Yang, Xiaoxu Xie

**Affiliations:** Faculty of Engineering, China University of Geosciences, Wuhan 430074, China

**Keywords:** landslide, high cut-slope, weak bedding plane, fault, extreme rainfall

## Abstract

High cut-slopes are widespread in engineering constructions and often converted into landslides. Some extreme circumstances facilitate the landslide process, such as the weak bedding plane, rainfall, and faults. Therefore, this paper intends to offer insights into the influence of the weak bedding plane, extreme rainfall as well as faults on the landslide process of the high cut-slope. In this paper, the Anling landslide in Anhui Province, China, is selected as an example. Geological surveys, displacement monitoring, data analysis, as well as numerical simulation are carried out. The entire excavation construction and landslide deformation process are simulated to reveal the formation mechanism of the landslide using the finite difference code, FLAC3D. The effects of the fault on the landslide and the effectiveness of adjusting protection measures by adding piles are investigated on the basis of the finite difference analysis. According to monitoring data and numerical simulation, the weak bedding plane and extreme rainfall are considered the main factors leading to the Anling landslide. Field investigation and numerical experiments indicate that the fault shall facilitate and accelerate the landslide process. The construction of piles in a suitable position for the landslide is a reasonable and economical measure to stabilize the landslide.

## 1. Introduction

High cut-slopes are widespread in engineering constructions to meet the requirement of terrain modification [1,2,3]. Because of the proximity, uncontrolled deformations of high cut-slopes have great effects on the service performance of buildings or transportation infrastructure [4,5]. Under extreme conditions, such as rainfall or earthquake, high cut-slopes usually become unstable, causing great damage to human lives and properties [6,7,8,9]. Thus, it is of great significance to ascertain the failure mechanism of high cut-slope so that prevention and mitigation strategies can be implemented to maintain the performance of infrastructure at an affordable cost.

Dip-slope landslide events of high cut-slopes often occur under certain circumstances, depending on inherent geological characteristics as well as external triggering factors [10,11,12]. In general, the inherent geological characteristics consist of lithological characteristics and structural characteristics. A weak bedding plane is one of the most noticeable aspects of lithological characteristics and tends to adversely affect slope stability due to its lower mechanical properties [13,14]. For example, shale and mudstone are the most representative weak rocks. The shales often present softening, fissuring, and dilatant behavior, leading to long-term progressive slope failures [15]. Under the influence of weathering process and water weakening, the progressive slope failure could convert into a catastrophic landslide [16]. The chemical water–rock interaction of black shale and its deterioration process also controls the development and behavior of slip zones of landslides [17]. Besides, the fault also plays an important role in the formation of landslides as a widespread geological structure [18,19]. A fault usually develops into a slip surface or facilitates the development of the main scarp for a translational landslide [20]. Rock slopes in northern Norway show the rupture at the head of the slope utilizing the fault plane [21]. One of the reasons for the phenomenon is the reduction in the strength of the fault zone and the rock surrounding it, which is the result of increasing fracture density around fault zones [22]. Therefore, the influence of the damage zone associated with fault on landslides cannot be negligible. Moreover, the variation of external factors is equally able to trigger landslides, especially heavy rainfall [23,24,25,26]. Generally, rainfall controls the landslide process in several ways. Rainfall infiltration increases the density of material and pore pressure, leading to unbalance between driving and resisting forces [27,28]. Another influence of rainfall on slope stability is the decrease in the rock strength because of the effects of water-induced weakening, and laboratory tests also indicate that the strength of rock will greatly decrease when saturated [29]. Numerous cases indicate that the relation between precipitation and landslide movement is obvious [30,31,32].

The failure mechanism of high cut-slope becomes quite complicated when complex geological conditions and extreme rainfall exist simultaneously. A dip-slope landslide event of the high cut-slope, named Anling landslide, was chosen as an example, which is located in Anhui Province, China. This paper attempts to reveal the influence of weak bedding planes, fault as well as heavy rainfall on the landslide process of the high cut-slope. Then, geological surveys, displacement monitoring, and numerical simulation were carried out. Some interesting conclusions were obtained about the effects of weak bedding planes, fault, and heavy rainfall on the landslide. In order to prevent further slides, recommendations for landslide protection were also discussed at the end.

## 2. Study Area

The study area is located in the eastern part of China and belongs to Qimen County, Anhui Province (Figure 1a). The position of the study area is 117°38′33″ N and 30°6′47″ E. A highway under construction passes through the study area, and a landslide, named Anling landslide, occurred during the construction (Figure 1b). Field surveys, drilling, and surface monitoring are adopted to investigate the landslide. The topographic map of the study area is shown in Figure 1c. Although protection measures were utilized to protect the high cut-slope, landslide deformation occurred during the period of construction due to heavy rainfall. Because of the landslide occurrence and deformation, the engineering design was modified three times, leading to extra costs in highway construction.

The Anling landslide covers an area of 2.3 × 10^4^ m^2^, with a maximum width of 170 m, a longitudinal length of 220 m, and a maximum thickness of 30 m. The landslide volume is about 4.6 × 10^5^ m^3^. The sliding direction is 73° with an average slope angle of approximately 35°. In terms of geological characteristics, the landslide involved the lower Silurian Xiaxiang formation with siltstone and shale. The landslide mass is mainly made up of highly weathered and well-jointed siltstone. The sliding surface is composed of moderately weathered shale with weak mechanical properties, and the bedrock is dominated by moderately weathered siltstone (Figure 2). In addition, a fault of 1.5 m width was observed at the toe of the landslide (Figure 3a). The presence of a fault and weak bedding plane is important geological factors causing the landslide.

## 3. Deformation Characteristics of the Anling Landslide

Anling landslide, which is affected by excavation, weak bedding plane, fault, as well as rainfall, generates deformation associated with these factors. During some excavation stages and rainfall periods, large deformation of the landslide occurred, and the deformation is further intensified due to weak, weathered, and well-jointed rock. In this study, the fault increases the degree of rock fragmentation. In the section, the process of slope excavation is firstly introduced, including slope excavation sequence and protection measures. Then landslide deformation process, fault characteristics, and monitoring data are presented.

### 3.1. Process of the Slope Excavation

The design of the slope is a 5-grade excavated slope with a maximum height of up to 38 m (Figure 3b). The slope ratios of the 1st and 2nd are 1:1, and the slope ratios of the 3rd and 4th are 1:0.75, while the slope ratio of the 5th-grade slope is 1:0.4. The height of the 1st to the 4th-grade slope is 8 m, and the height of the 5th-grade slope is 6 m. The 2nd and 3rd-grade slopes are strengthened with three rows of prestressed anchor lattice beams respectively. The length of the anchor is 25 m and 20 m. Three rows of non-prestressed anchor lattice beam and one row of prestressed anchor lattice beam were used on the 4th-grade slope. Due to the occurrence of large deformation on the slope, the protection measure conducted on the 5th-grade slope is the most complicated, composed of one row of non-prestressed anchor lattice beam, and four rows of prestressed anchor lattice beam. The process of slope excavation and the effect of heavy rainfall on the slope is divided into six stages (Figure 3b). The excavation was carried out during stages 1 to 5, while extreme rainfall events happened during the construction of the sixth stage.

The slope excavation began in September 2019. There was no identifiable deformation from stage-1 to stage-4, and the 1st to 4th-grade slopes excavation as well as the protection measures were completed successfully. Only three rows of non-prestressed anchors (three yellow lines shown on the 4th-grade slope in Figure 3b) were used on the 4th-grade slope during this period. However, when the 5th-grade slope was excavated in August 2020, the failure occurred at the upper benches and slopes. Rapid remedial work was carried out and the excavation area of the 5th-grade slope was backfilled. Subsequently, the original design was modified to add one row of prestressed anchors on the 4th-grade slope. Then, the 5th-grade slope was excavated again, followed by the construction of two rows of prestressed anchors and one row of non-prestressed anchors (three orange lines on the 5th-grade slope are shown in Figure 3b).

In early July 2021, there was extreme rainfall that lasted for five days with 100 mm/day, resulting in further landslide movement. Several cracks were observed on the upslope of the Anling landslide, and large deformation was found at the toe of the landslide, especially the failure of the 5th-grade slope. For the sake of stabilizing the slope, the construction design was modified again. Two additional rows of prestressed anchors (two red lines on the 5th-grade slope are shown in Figure 3b) were conducted on the 5th-grade slope. Due to extra prestressed anchors construction, landslide movement slowed down so that the landslide seems to become stable.

### 3.2. Deformation Characteristics of the Anling Landslide

Landslide deformation adversely affected highway construction. In order to learn about landslide deformation, six displacement monitors were installed on the slopes (Figure 1c). Each longitudinal section has three displacement monitors, located on the 2nd, 3rd, and 4th-grade slopes respectively. There were eight cracks distributed on the upslope (Figure 1c), indicating the potential for landslide failure. A large mass movement event took place at the toe of the landslide, causing the failure of the lattice beam on the 5th-grade slope. Moreover, cracks of different lengths and widths were observed on each bench although protection measures were carried out. According to the field investigation and displacement monitoring, these deformations occurred during the construction of stage-5 and stage-6.

Landslide deformation appeared after the first excavation at stage 5, and cracks were observed on the 5th-grade slope as well as the upper benches. A crack of 15 m in length was discovered on the 5th-grade slope (Figure 4a). The crack developed along the boundary between moderately weathered shale and highly weathered siltstone. Along the third bench, there was a crack of 35 m in length, which was parallel to the extension direction of the bench (Figure 4b). In addition, two cracks perpendicular to the extension direction of the second and first bench were 2 m and 1.5 m in length (Figure 4c,d). The interception drain on the head of the high-cut slope was also cracked (Figure 4e). Besides, a fault of 1.5 m width was revealed on the 5th-grade slope during the first excavation (Figure 3a). The fault developed in the highly weathered siltstone, causing further fracture of rock. Due to the fault being initially hidden beneath the surface, the construction design ignored the impact of the fault.

Extreme rainfall lasted for five days during stage 6, leading to large deformation of the Anling landslide. The 5th-grade slope and its lattice beam suffered extensive damage. Lattice beams at the head of the 5th-grade slope were destructive and slipped towards the highway (Figure 5a). In the middle of the 5th-grade slope, cracks appeared at the lattice beam due to landslide movement (Figure 5b). Furthermore, the loosening of rock mass and empty areas at the toe of the 5th-grade slope suggested the position of the potential slip surface (Figure 5c). There were eight cracks distributed on the middle and back of the landslide (Figure 1c), with a width of 0.3–10.5 cm (Figure 5d–k). Considering the spatial distribution, a tendency could be found that the width of the crack decreased with the distance from the toe of the landslide.

Displacement monitoring data from December 2020 to October 2021 was obtained, involving the latter part of stage 5 as well as the entire stage 6 (Figure 6). Unfortunately, displacement monitor-6 was broken down on 20 June 2021, resulting in subsequent missing data. The step-like overall horizontal displacement time series suggests an intermittent movement of the landslide. As shown in Figure 6, landslide movement is associated with construction and rainfall. The landslide movement appeared during the building of the anchor lattice beam on the 5th-grade slope at stage 5, rainfall, and the building of an extra prestressed anchor at stage 6. Because deformations are unavoidable during construction disturbances, the displacements associated with rainfall deserve more attention. The maximum landside displacement reached 57 mm on the 4th-grade slope and the minimum was 25 mm on the 2nd-grade slope. Apparently, the landslide displacement decreased with the distance from the toe of the landslide. Furthermore, the displacement monitors in section 2-2′ detected larger displacements than those in section 1-1′. After the extra prestressed anchor was completed, the displacement time series remained horizontal, and the landslide can be regarded as a stable state.

## 4. Numerical Simulation of the Anling Landslide

In order to investigate the landslide process of the high cut-slope, a two-dimensional numerical simulation was carried out on the Anling landslide using FLAC3D 6.0 (ITASCA, Minneapolis, MN, USA). Taking into account the movement of the entire landslide as well as the integrity of the construction process, section 1-1′ was selected to simulate landslide deformation from stage 1 to stage 6. In the section, the constitutive model and parameters of numerical simulation were presented first. Next, the numerical simulation result of landslide deformation was described. Finally, the numerical simulation result along with monitoring data and field investigation were employed to analyze the failure mechanism of the Anling landslide.

### 4.1. Numerical Model Descriptions

Based on the field investigation, the Anling landslide is composed of three geotechnical materials, containing highly weathered siltstone, moderately weathered shale, and moderately weathered siltstone. An elastic-plastic constitutive model with the Mohr-Coulomb criterion is adopted in this study to describe the mechanical behavior of the three geotechnical materials. The numerical model follows the real sequence of construction, including excavation, protection measures, and rainfall. Here, the excavation period was simulated by removing the regions (Figure 7a), whereas anchors and lattice beams are modeled by cable and beam structure elements in FLAC3D 6.0. In addition, to depict infiltration during the extreme rainfall event, the upper boundary of the model is regarded as an infiltration boundary, while the left, right, and lower boundaries are served as impermeable boundaries (Figure 7b). Since no groundwater was observed during the field investigation, the groundwater can be negligible in the model.

Material parameters utilized in the numerical model are presented in Table 1, obtained from literature and inversion [9,33,34,35,36]. Mechanical properties such as cohesion, internal friction angle, and Young’s modulus of moderately weathered shale are reduced by water-induced weakening [37,38,39]. Xu et al. [40] report that the cohesion and Young’s modulus of shale decreased by 50% and 25% respectively after the five-day water immersion. Bian et al. [41] considered that the internal friction angle of shale shall drop to 93% after the ten-day water immersion. Taking into account the weathering and fracture of the shale, the cohesion, internal friction angle, and Young’s modulus of the shale after the five-day water immersion was determined to be 47%, 85%, and 75% of the original values (Table 1). Moreover, records of rainfall in the study area show that the average annual rainfall is 1700 mm, with a minimum of 890 mm (1978) and a maximum of 2500 mm (1983). The distribution of rainfall throughout the year illustrates 60% of the rainfall between May and August. The maximum recorded 24-h was 298 mm (1983) and the maximum one-hour recorded was 89 mm for the same year. In order to reflect the extreme rainfall condition, the precipitation in the model is considered as 100 mm/day, accumulatively 500 mm for five consecutive days.

### 4.2. Result of Numerical Simulation

The sequence of construction was simulated step-by-step, as shown in Figure 8. Deformation of the Anling landslide is infinitesimal before stage 5 (Figure 8a), and the maximum horizontal displacement is only 1.50 mm. Subsequently, Figure 8b illustrates the simulated deformation due to the first excavation at stage 5. The landslide moves along the interface between highly weathered siltstone and moderately weathered shale, then runs out at the toe of the 5th-grade slope. The maximum horizontal displacement is 33.33 mm, and landslide displacement decreases with the distance from the toe, which is in good agreement with the failure phenomenon that occurred in August 2020. Slight deformation can be observed after backfilling as well as prestressed anchors are simulated (Figure 8c). The same situation arises for the second excavation at stage 5, with a maximum horizontal displacement of less than 0.59 mm (Figure 8d).

During the period of extreme rainfall, significant changes in the saturation of materials can also be observed in the model. The saturation increases near the interface between the highly weathered siltstone and moderately weathered shale, while zero saturation is maintained in other regions. Saturated regions appear around the interface, expanding as the rainfall progresses. Most areas around the interface are saturated after five days of rainfall. Because the moderately weathered shale has been soaked in water for five days, a reduction of the shale strength and subsequent deformation is simulated in the model (Figure 8e). Besides, there is a nearly planar slip surface between the highly weathered siltstone and moderately weathered shale. The sliding rock mass moves along the slip surface as a sliding block, and rotation occurs at the toe. The maximum horizontal displacement is 59.10 mm on the 5th-grade slope, and the horizontal displacement of the 4th-grade slope is from 49.50 mm to 55.00 mm, which is consistent with the field investigation and displacement monitoring data. After the construction of an additional prestressed anchor at stage 6, the landslide became stabilized (Figure 8f).

### 4.3. Failure Mechanism of the Anling Landslide

Based on the field investigation, monitoring data, as well as numerical simulation, the failure mechanism of the Anling landslide can be analyzed. The inherent geological reason is the presence of weak rock formations. In addition, the mechanical properties of moderately weathered shale are lower than those of other rocks on the slope, resulting in a potential slip surface. The excavation also disrupts the balance between driving and resisting forces, triggering rock mass movement along the potential sliding surface (Figure 4a and Figure 8b). Moreover, extreme rainfall is the most important triggering factor, inducing an increase in the deformation (Figure 5). Precipitation infiltrates rapidly through the highly weathered siltstone to the moderately weathered shale because the well-jointed siltstone has a high permeability coefficient. However, low permeability shale results in saturation around the interface between the highly weathered siltstone and the moderately weathered shale. With the progress of rainfall, prolonged soaking shall lead to the weakening of the shale’s mechanical properties, then large landslide displacement takes place (Figure 6). The numerical simulation illustrates that the rock mass slides along the potential slip surface composed of water-weakened shale (Figure 8e). Overall, the fragile bedding plane and extreme rainfall are the main reasons leading to the deformation and failure.

## 5. Discussion

Besides the weak bedding plane and extreme rainfall, the fault also plays an important role in the deformation and failure of the Anling landslide. Although the development of the fault is localized, we believe that it promotes the occurrence of landslides. Monitoring data and numerical simulation are utilized to discuss the effect of the fault on the landslide. Because of the complex landslide deformation mechanism and intermittent movement, the design of landslide protective measures was revised three times, causing delays and extra costs in highway construction. Thus, the selection of appropriate landslide protection measures shall be discussed to avoid the same circumstance.

### 5.1. Effect of the Fault on the Anling Landslide

The fault located at the toe of the landslide can only be described in section 2-2′ (Figure 2). According to the monitoring data, the displacement monitors in section 2-2′ detected greater displacement than section 1-1′, indicating that the localized fault causes an increase in the landslide displacement. Fractured rock mass in a fault damage zone has lower mechanical properties than rock mass in non-damage zones [42,43], which might explain the inhomogeneous deformation of the Anling landslide.

In order to verify the hypothesis, a two-dimensional numerical simulation was carried out using FLAC3D 6.0. Section 2-2′ was selected to simulate landslide deformation affected by the fault structure during rainfall. Taking into account the length of the fault and the fragmentation of the surrounding rock mass, the damage zone was modeled as a rectangular shape with a size of 21 m × 23 m centered on the fault (Figure 9a). The rock strength of the damage zone shows a decreasing trend compared to the non-damage zone [44]. Yin et al. [45] found that the strength and deformation parameters of the fault fracture rock mass are decreased by 19.20–86.51%, including cohesion, internal friction angle, and Young’s modulus. According to the field investigation and numerical inversion, the cohesion, internal friction angle, and Young’s modulus of the damage zone were assumed to have a reduction of 20% compared to the non-damage zones in this study (Table 1), while other parameters and boundaries were the same as the previous model of section 1-1′.

The simulated landslide deformation affected by the fault structure is shown in Figure 9b. The horizontal displacement of the 4th-grade slope is from 56.00 mm to 70.00 mm, which is consistent with displacement monitoring data. The maximum horizontal displacement is 121.30 mm on the 5th-grade slope, and the landslide displacement is greater than that of the previous simulation of section 1-1′. Therefore, the results in accordance with the field investigation confirm that the conjecture in this paper is reasonable. The localized fault causes a decrease in the mechanical properties of the surrounding rock mass, resulting in greater landslide deformation under unfavorable conditions. This process facilitates and accelerates the failure of the Anling landslide.

### 5.2. Selection of Appropriate Landslide Protection Measures during Construction

The complex landslide deformation mechanism makes it difficult to select appropriate protection measures. Besides, due to the limited information obtained from the pre-construction geological survey, the original protection strategy for landslides has major limitations and failure is almost inevitable. However, the great landslide deformation during rainfall is able to be avoided by adopting appropriate protection measures based on secondary geological exploration. Therefore, we then discuss how to select the protection measure to prevent the second movement of the Anling landslide.

In addition to the anchor lattice beam used in the construction, a pile is a common and useful landslide protection measure. In this study, we attempt to adjust the protection measure by adding piles and evaluate the effectiveness of the stabilization method by two-dimensional numerical simulation. Section 1-1′ was selected to simulate the second landslide deformation after adding piles. The dimensions of pile row 1 and pile row 2 are 2 × 3 × 62 m and 2 × 3 × 46 m (width × height × length) respectively. In order to prevent damage to anchors, piles are placed in the middle and upper parts of the landslide. Piles are simulated by pile structure elements in FLAC3D 6.0, and the material parameters are listed in Table 1.

Landslide deformation under different protection strategies is shown in Figure 10. Compared with the original protection strategy, the modeled displacement of adding pile row 1 is extremely small. The maximum horizontal displacement is reduced from 59.10 mm to 4.10 mm (Figure 10b). The result suggests that the piles can effectively prevent the deformation process of the landslide corresponding to rainfall. Moreover, the landslide displacement shall be further reduced when pile row 1 and pile row 2 are modeled simultaneously, with a maximum horizontal displacement of 2.90 mm (Figure 10c). However, given that the cost of this option reaches almost doubled, such a little improvement in landslide stability seems worthless. Therefore, taking into account the construction cost and protection effect, adding one row of piles in the middle of the landslide is a reasonable protection measure to avoid the second movement under the circumstance that anchors have been embedded in the toe of the landslide.

## 6. Conclusions

The landslide process of a high cut slope during highway construction was investigated in this study, which is affected by weak bedding planes, faults, and extreme rainfall. Based on a refined geological survey, displacement monitoring, data analysis, as well as numerical simulation, the main conclusions are summarized as follows:The weak bedding plane resulting in a potential slip surface is the inherent geological reason for the landslide, while the hydro-weakening of moderately weathered shale due to extreme rainfall is the main external triggering factor for the landslide. The combined effect of the two main factors induced the landslide process of the high-cut slope.The fault causes a decrease in the mechanical properties of the surrounding rock mass, which is conducive to landslide deformation. The process shall facilitate and accelerate the failure of the landslide.Arranging and installing one row of piles in the middle of the landslide is a reliable and economical protection measure to prevent the landslide from deformation when the anchors have been embedded in the toe of the landslide.

The insights gained from our study contribute to learning about the landslide process of a high cut-slope and the formation mechanism of landslides under complicated conditions, especially during highway construction. The landslide protection recommendations in this paper can provide a reference for local government and/or community decision-makers.

## Figures and Tables

**Figure 1 sensors-22-06790-f001:**
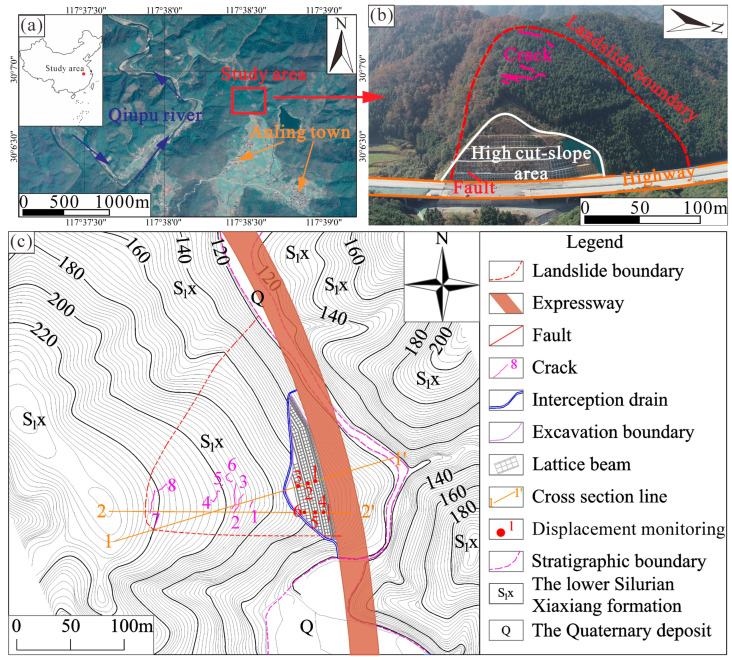
Location map of study area. (**a**) Geographic position of study area. (**b**) Fault, high cut-slope area, and deformation of the Anling landslide marked on an aerial photograph. (**c**) Topographic map of the Anling landslide.

**Figure 2 sensors-22-06790-f002:**
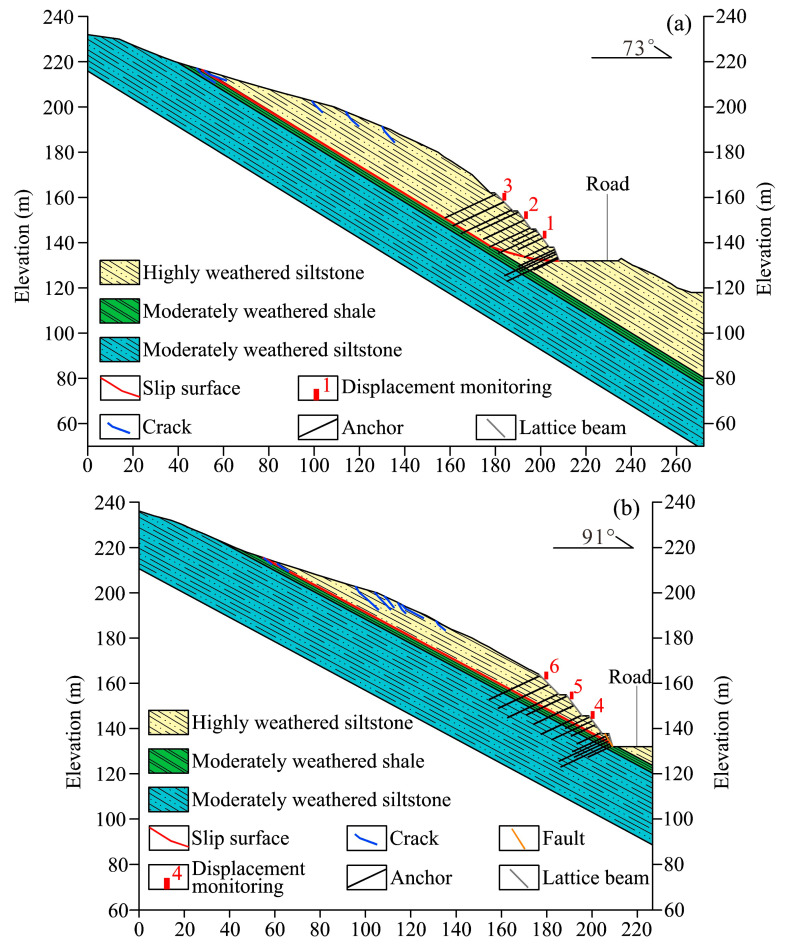
Longitudinal section of the Anling landslide. (**a**) 1-1′ section. (**b**) 2-2′section.

**Figure 3 sensors-22-06790-f003:**
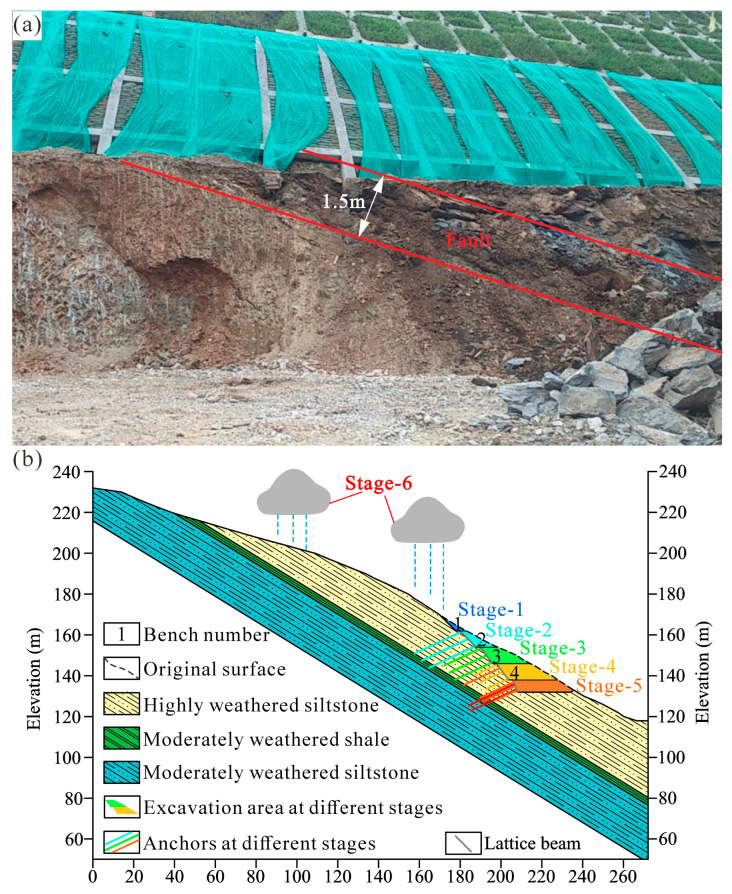
Fault characteristics of the Anling landslide and process of the slope excavation. (**a**) The photo of the fault. (**b**) The different stages of slope excavation.

**Figure 4 sensors-22-06790-f004:**
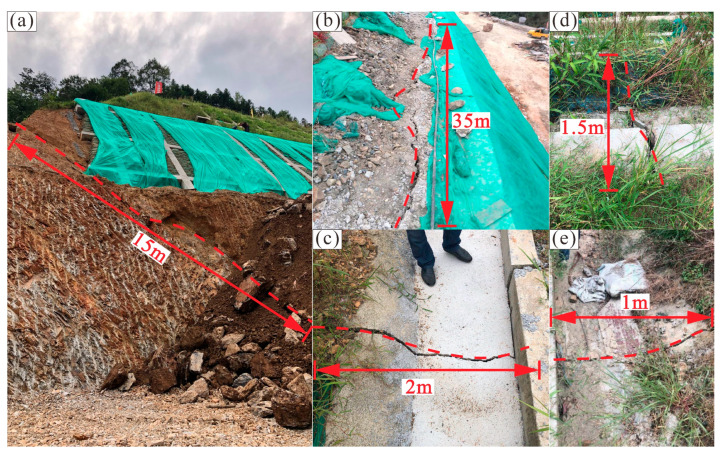
Photos of the Anling landslide deformation at stage 5. (**a**) Crack on the 5th-grade slope. (**b**) Crack on the third bench. (**c**) Crack on the second bench. (**d**) Crack on the first bench. (**e**) Crack on the interception drain of the high-cut slope.

**Figure 5 sensors-22-06790-f005:**
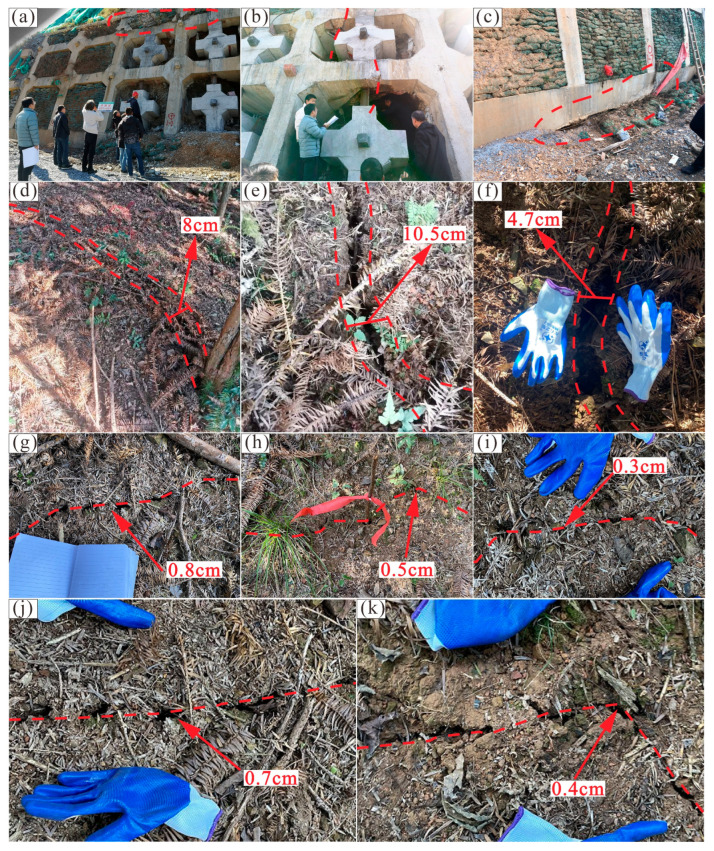
Photos of the Anling landslide deformation at stage 6. (**a**) Failure of lattice beam at the head of the 5th-grade slope. (**b**) Crack of lattice beam at the middle of the 5th-grade slope. (**c**) Deformation of slope protection at the toe of the 5th-grade slope. (**d**) Width of Crack1. (**e**) Width of Crack2. (**f**) Width of Crack3. (**g**) Width of Crack4. (**h**) Width of Crack5. (**i**) Width of Crack6. (**j**) Width of Crack7. (**k**) Width of Crack8.

**Figure 6 sensors-22-06790-f006:**
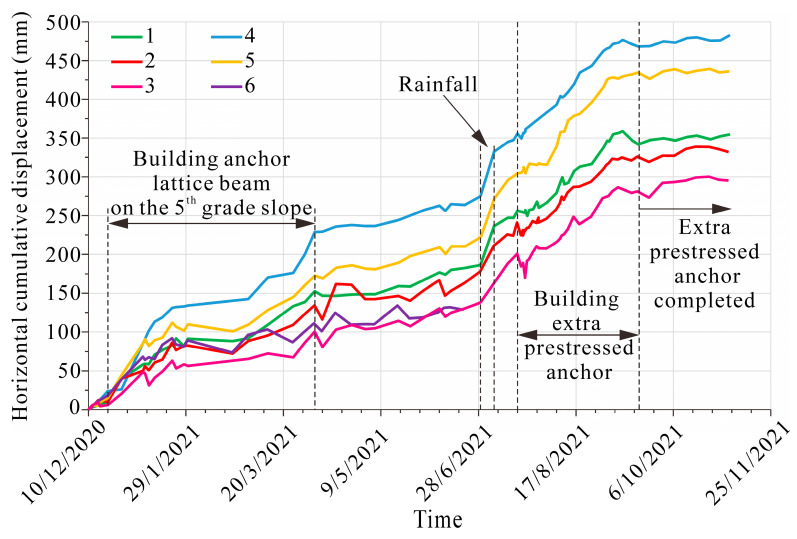
Horizontal cumulative displacement of the Anling landslide.

**Figure 7 sensors-22-06790-f007:**
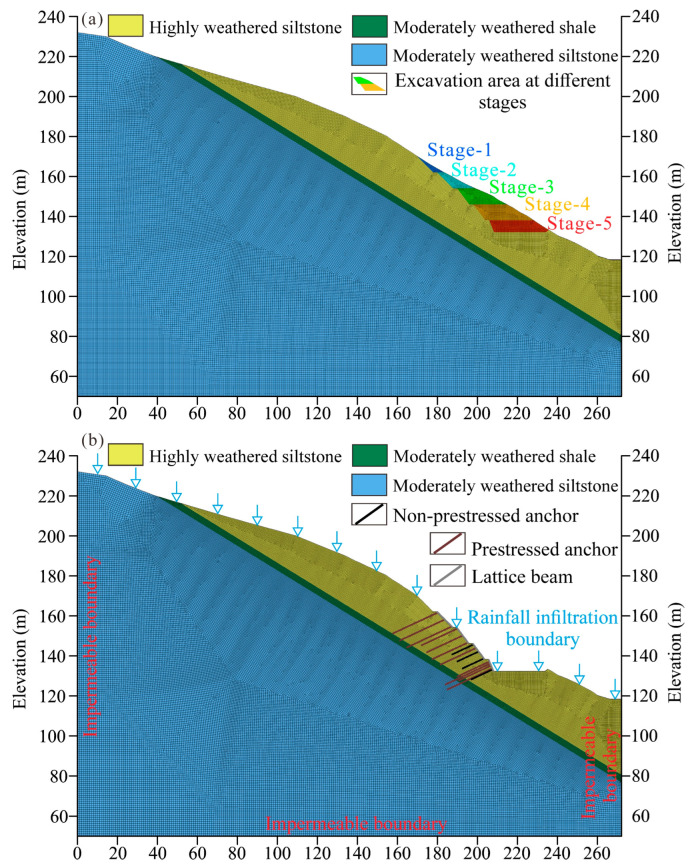
2D numerical model of section 1-1′. (**a**) Numerical model under natural conditions during construction. (**b**) Numerical model under extreme rainfall conditions.

**Figure 8 sensors-22-06790-f008:**
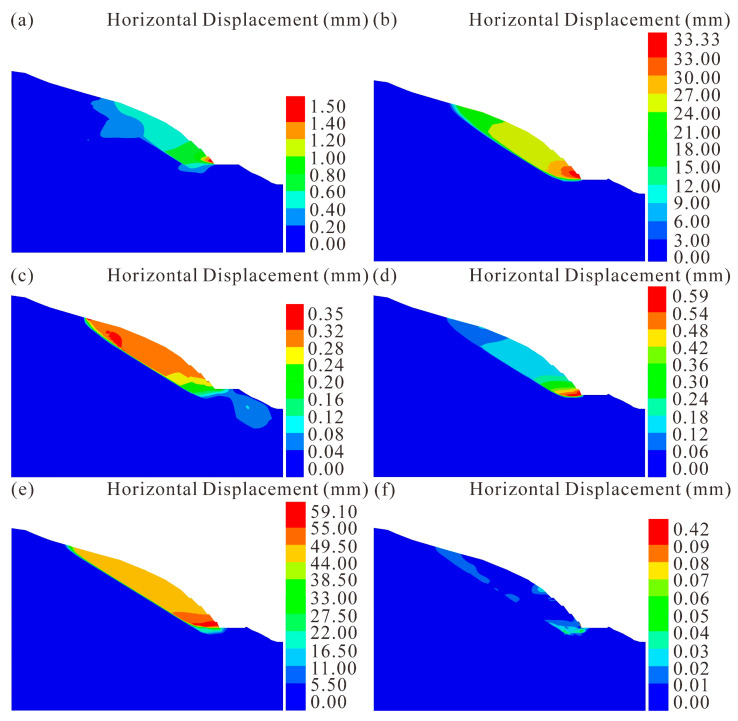
Numerical simulation of the Anling landslide. (**a**) Deformation of the landslide before stage 5. (**b**) Deformation of the landslide after the first excavation at stage 5. (**c**) Deformation of the landslide after backfilling and building prestressed anchor at stage 5. (**d**) Deformation of the landslide after the second excavation at stage 5. (**e**) Deformation of the landslide during rainfall at stage 6. (**f**) Deformation of the landslide after building extra prestressed anchor at stage 6.

**Figure 9 sensors-22-06790-f009:**
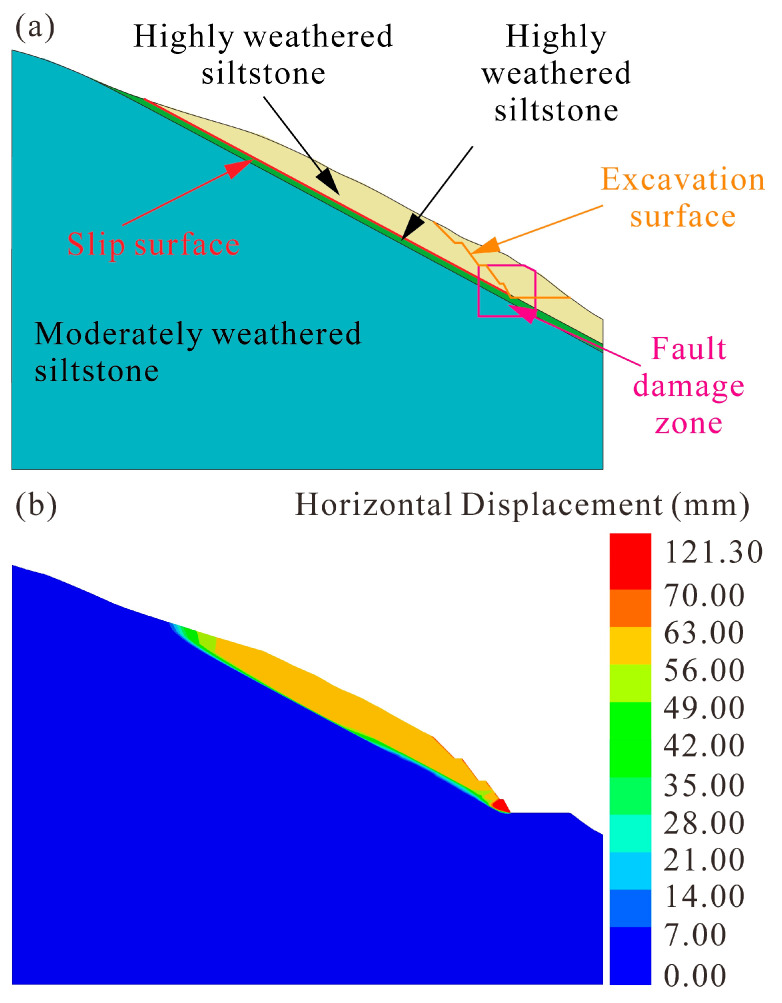
The influence of fault on landslide deformation. (**a**) Fault damage zone on 2-2′ section. (**b**) Numerical simulation of landslide deformation under the fault-influenced condition.

**Figure 10 sensors-22-06790-f010:**
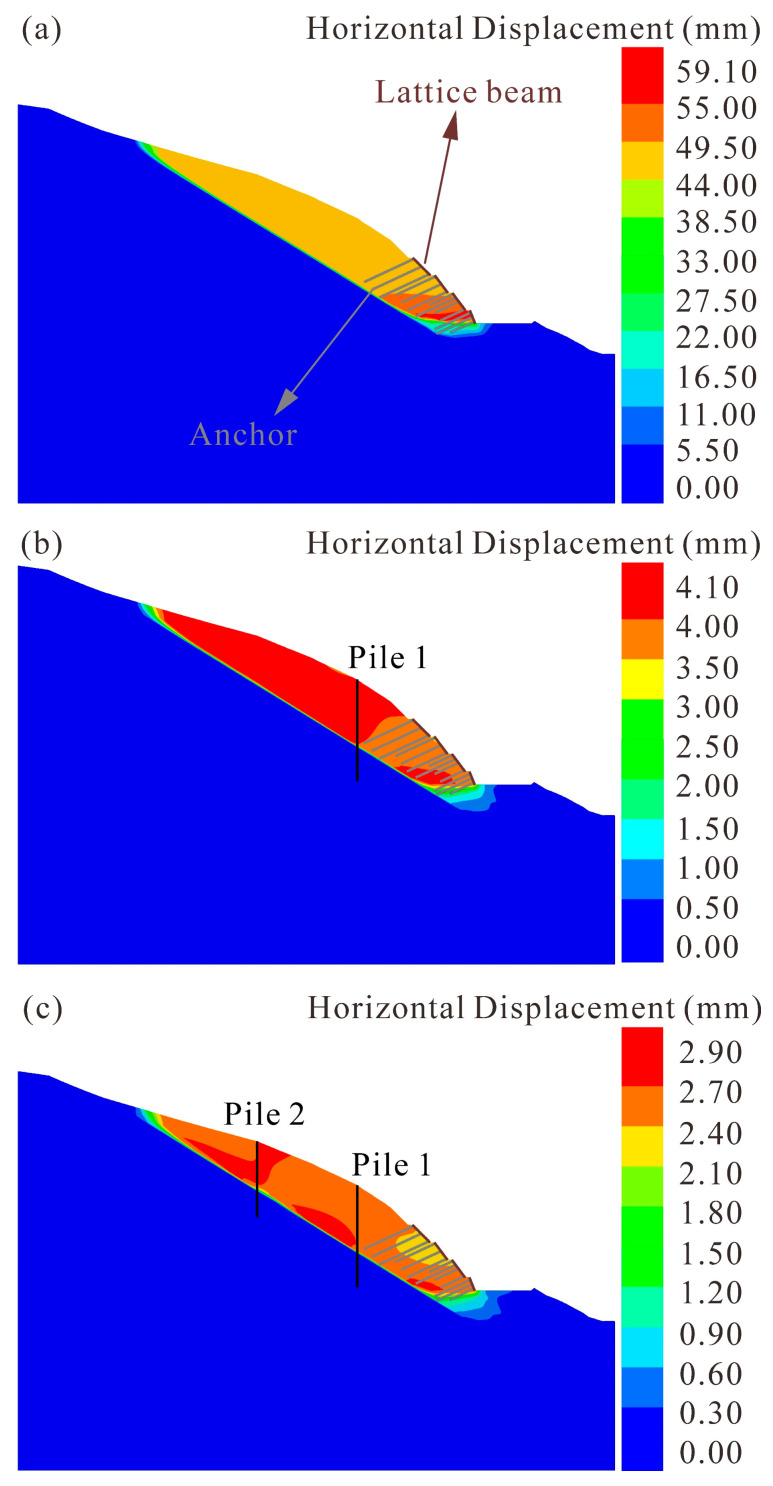
Numerical simulation of different protection strategies for the landslide. (**a**) Original protection strategy. (**b**) One row of piles in the middle of the landslide. (**c**) Two rows of piles in the middle and upper parts of the landslide respectively.

**Table 1 sensors-22-06790-t001:** Material properties used in the numerical model.

Material	Density (kg/m^3^)	Cohesion (kPa)	Internal Friction Angle (°)	Young’s Modulus (MPa)	Poisson’s Ratio	Permeability Coefficient (cm/s)	Porosity	Initial Force (kN)
Highly weathered siltstone	2150	38	30	5800	0.33	1.0 × 10^−4^	0.3	-
Moderately weathered shale	2250	43	20	2000	0.29	1.0 × 10^−9^	0.1	-
Moderately weathered siltstone	2350	40	35	9100	0.19	3.0 × 10^−7^	0.1	-
Weakening moderately weathered shale	2250	20	17	1500	0.29	1.0 × 10^−9^	0.1	-
Highly weathered siltstone in damage zone	2150	30.4	24	4640	0.33	1.0 × 10^−4^	0.3	-
Moderately weathered shale in damage zone	2250	34.4	16	1600	0.29	1.0 × 10^−9^	0.1	-
Moderately weathered siltstone in damage zone	2350	32	28	7280	0.19	3.0 × 10^−7^	0.1	-
Weakening moderately weathered shale in damage zone	2250	16	13.6	1200	0.29	1.0 × 10^−9^	0.1	-
Non-prestressed anchor	-	-	-	200,000	-	-	-	-
Prestressed anchor	-	-	-	195,000	-	-	-	400
Lattice beam	-	-	-	30,000	0.25	-	-	-
Pile	-	-	-	30,000	0.25	-	-	-

## Data Availability

Not applicable.

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
