# Peer review of "Effects of Weak Bedding Plane, Fault, and Extreme Rainfall on the Landslide Event of a High Cut-Slope"

_sensors, 2022, doi:10.3390/s22186790_

Round 1

Reviewer 1 Report

In this paper, the authors study the transformation from a high cut-slope in highway construction to a landslide under adversely conditions and investigated the influences of weak bedding plane, fault, and extreme rainfall on this process. Field survey, monitoring data and numerical simulation are utilized to analyze the landslide mechanism. A little work is required before the manuscript can be considered for publication. Meanwhile, I consider needs a minor revision. Here are my comments:

(1)   The authors study the transformation from a high cut-slope in highway construction to a landslide under adversely conditions and investigated the influences of weak bedding plane, fault, and extreme rainfall on this process. If you can further analyze their relationship in the introduction based on previous achievements, it may be better.

(2)   It is suggested to supplement the location of the study area in China.

(3)   The numbers on the Figure 4 and Figure 5 are not clear, and need to add white background to the numbers.

(4)   To avoid misunderstanding, the caption of the figures should be consistent with the text. The caption of the Figure 4e should be changed to “Crack on the interception drain of the high-cut slope.” The caption of the Figure 5a should be changed to “Failure of lattice beam at the head of the 5th grade slope.”

(5)   There are a few grammar errors in this paper. For examples, “extra costs” should replace “extra cost” on line 92; “stabilizing” should replace “stabilize” on line 164.

(6)   The serial numbers of different excavation stages need to be marked on the Figure 7a.

(7)   The description of the cohesion reduction in the text (50% of the original values, line 285) is inconsistent with the value calculated from the data given in the Table 1(47% of the original values, page 11). The data in the Table 1 and description in the text need to be checked.

(8)   To avoid misunderstanding, the mark “Area influenced by fault” in the Figure9a should be changed to “Fault damage zone”, and the caption of the Figure9a should be also changed to “Fault damage zone on 2-2’ section.” The caption of the Figure 10c should be changed to “Two rows of piles in the middle and rear of the landslide respectively.”

Author Response

Thank you very much for the suggestions and comments. Please see the attachment.

Reviewer 2 Report

1.     The term ‘transformation’ is seldom used in landslide studies. I suggest you change title to ‘Effects of Weak Bedding Plane, Fault, and Extreme Rainfall on the landslide event of a high cut-slope’ for your reference. You may consider to use “progressive failure” or other terms instead.

2.     In the main text you should define the term ‘transform’ first if you keep using it in the whole text.

3.     I suggest us ‘dip-slope landslides’ instead of ‘bedding rock landslides’.

4.     Figure 2: The colors of the 'Slip surface', 'Anchor' and 'Crack' are too close, not easy to distinguish.

5.     Line 231: ‘Because the deformations are unavoidable during construction disturbances, …’

6.     Figure 7 shows ‘2D numerical model of section 1-1’. So, your analysis was actually performed in two-dimensional, not three-dimensional, even though you used FLAC3D code, right? If so, you should let the reads know.

7.     Most of the material properties used in the numerical model are reasonable (Table 1); however, I think the permeability coefficient of the weathered siltstone is too high. You used 10 cm/s.

8.     I suggest to use ‘toe’ instead of ‘leading edge’.

9.     Line 434: Use ‘upper parts’ instead of ‘rear’.

10.  I made suggestions and comments in the attached PDF file for your reference.

Author Response

Thank you very much for the suggestions and comments. We have tried our best to revise the manuscript according to your kind and construction comments and suggestions. Please see the attachment.
